# State of the Art on the Role of *Staphylococcus aureus* Extracellular Vesicles in the Pathogenesis of Atopic Dermatitis

**DOI:** 10.3390/microorganisms12030531

**Published:** 2024-03-06

**Authors:** Marina Passos Torrealba, Fabio Seiti Yamada Yoshikawa, Valeria Aoki, Maria Notomi Sato, Raquel Leão Orfali

**Affiliations:** 1Laboratory of Medical Investigation in Dermatology and Immunodeficiencies (LIM-56), Department of Dermatology, Faculdade de Medicina, Universidade de Sao Paulo, Sao Paulo 05403-000, SP, Brazil; marinatorrealba@usp.br (M.P.T.); valeria.aoki@gmail.com (V.A.); marisato@usp.br (M.N.S.); 2Division of Molecular Immunology, Medical Mycology Research Center, Chiba University, Chiba 260-8670, Japan; faseiti@chiba-u.jp

**Keywords:** *Staphylococcus aureus*, extracellular vesicles, atopic dermatitis, microbiota

## Abstract

Atopic dermatitis (AD) is a chronic and relapsing inflammatory cutaneous disease. The role of host defense and microbial virulence factors in *Staphylococcus aureus* skin colonization, infection, and inflammation perpetuation in AD remains an area of current research focus. Extracellular vesicles (EV) mediate cell-to-cell communication by transporting and delivering bioactive molecules, such as nucleic acids, proteins, and enzymes, to recipient cells. *Staphylococcus aureus* spontaneously secretes extracellular vesicles (SA-derived EVs)*,* which spread throughout the skin layers. Previous research has shown that SA-derived EVs from AD patients can trigger cytokine secretion in keratinocytes, shape the recruitment of neutrophils and monocytes, and induce inflammatory AD-type lesions in mouse models, in addition to their role as exogenous worsening factors for the disease. In this review article, we aim to examine the role of SA-derived EVs in AD physiopathology and its progression, highlighting the recent research in the field and exploring the potential crosstalk between the host and the microbiota.

## 1. Introduction

Atopic dermatitis (AD) is a chronic and relapsing inflammatory skin disease. It shows worldwide occurrence, affecting both children and adults, whose prevalence is approximately 15–20% and 1–5%, respectively [1,2,3]. Although a common disease, the pathogenesis of AD is a complex network of skin barrier defects, imbalance in adaptive and innate immunity, and chronic skin colonization by *Staphylococcus aureus* (*S. aureus*) [4,5,6], all of which drive an inflammatory vicious cycle culminating in the main symptom: pruritus [7,8,9,10]. But among the variables associated with AD development, the interaction between the host defense system and the microbial virulence factors involved in *S. aureus* skin colonization, infection and the perpetuation of inflammation are gathering increased attention [11], mostly due to clinical evidence that, in AD flares, decreased bacterial diversity associated with increased *S. aureus* abundance has been described and related to disease severity [11,12,13,14].

The virulence factors of *S. aureus* include the secretion of numerous exotoxins, comprising an assembly of polypeptides capable of injuring the host cell plasma membrane, the called pore-forming toxins: α-hemolysin and the bi-component leukocidins γ-hemolysin, the Panton Valentine leukocidin, LukED, and LukGH/AB, β-hemolysin (neutral sphingomyelinase), and the phenol soluble modulins (small amphipathic peptides), as well as staphylococcal enterotoxins (SEA to SEE, SEG to SEJ, SEL to SEQ and SER to SET), and 11 staphylococcal superantigen-like (SSL) toxins (SEIK to SEIQ, SEIU to SEIX) [15,16].

Almost all cell types release double-lipid membrane structures termed “extracellular vesicles (EVs)” into the extracellular lumen. EV production is a conserved biological phenomenon, considered a fundamental feature in all three domains of life: eukaryotic, prokaryotic and archaea cells [17,18,19]. In Eukaryotes, EVs are named based on their size, function, and biogenesis. Apoptotic bodies are very large vesicles with an average diameter of 1000–5000 nm, formed through the disassembly of the cell membrane. Ectosomes, also known as microvesicles, exhibit 100 to 1000 nm of diameter and are formed by the shedding of the plasma membrane. Exosomes are smaller vesicles ranging from 40 to 150 nm and originating from endosomal generation. Despite this categorization, the International Society for Extracellular Vesicles (ISEV) recommends the use of “extracellular vesicles” as a communal term for “particles naturally released from the cell that are delimited by a lipid bilayer and cannot replicate” [20,21,22]. These spherical structures play an essential role in cell communication by acting as biological carriers between different cells and tissues. Moreover, surface molecules on EVs can target specific cells and tissues, providing specificity to the recipient cell. The cargo includes acid nucleic molecules, such as DNA and RNA, proteins, lipids, enzymes, and any other molecules found in the donor cell. 

EVs exert a physiological role in cellular communication but have also been found to be important drivers in the pathophysiology of many diseases, including AD in recent years. Thus, the mechanism of action by which microbe-derived EVs may contribute to immune modulation, the delivery of virulence factors, the enhancement of antibiotic resistance, as well as the facilitation of biofilm production, thereby impacting the physiopathology of inflammatory diseases [20], are still being uncovered. The aim of this review is to examine the role of *S. aureus*-derived EVs (SA-derived EVs) in the pathophysiology and progression of AD.

## 2. Review Strategy

The authors performed a review in the PubMed, Embase, Google Scholar, and Web of Science databases. The search terms were: (‘staphylococcus aureus extracellular vesicles’ OR ((‘staphylococcus’/exp OR staphylococcus) AND aureus AND extracellular AND (‘vesicles’/exp OR vesicles)) OR ‘staphylococcus aureus exosomes’ OR ((‘staphylococcus’/exp OR staphylococcus) AND aureus AND (‘exosomes’/exp OR exosomes))) AND (‘atopic dermatitis’/exp OR ‘atopic dermatitis’ OR (atopic AND (‘dermatitis’/exp OR dermatitis))). The basis for assessment was the title and abstract for relevance to *S. aureus* EVs and AD. We included English-language articles published between 2009 and 2023. Figure 1 illustrates the accessed and selected papers. 

The search terms resulted in 11 articles in PubMed, 23 in Embase, 48 in Web of Science, and 572 in Google Scholar, but many with topics that did not match the purpose of this article. We excluded all articles that did not fit with the aim of the current review and accessed the relevant articles related to the topic. 

In Table 1, we present a timeline summarizing the main findings in the field uncovered during the last 15 years.

## 3. A Brief Background on SA-Derived EVs

Any cell has the potential to produce EVs. EVs released from Gram-negative bacteria range between 10–500 nm and were first described in 1966, named as outer membrane vesicles (OMVs) due to their origin from the bacterial outer membrane layer [37,38]. Curiously, the production of EVs by Gram-positive bacteria was only confirmed in 2009 [23], mostly due to a misconception; once they do not possess an outer membrane, they would be unable to liberate such structures [39]. Nonetheless, it was uncovered that Gram-positive bacteria-derived EVs are released from the single cytoplasmic cell membrane surrounded by a peptidoglycan-rich cell wall, and exhibit a diameter of 20–400 nm, and are therefore named membrane vesicles (MVs) [40]. In this review, however, we will keep the term “extracellular vesicles/EVs” regardless of the source organism.

*S. aureus* is also capable of producing and releasing EVs (here termed SA-derived EVs) that contain membrane proteins, as well as superantigens and enzymes, but which do not seem to display significant cytotoxicity after being internalized by the host cell [24,32]. In 2009, Lee et al. identified 90 proteins in *S. aureus* EVs through proteomic approaches, marking also the first observation of *S. aureus* spontaneously producing these structures [23]. More recently, another research group evaluated five different *S. aureus* strains obtained from various hosts, including humans, bovine, and ovine animals, and demonstrated the presence of a highly conserved EV core proteome for SA-derived EV [41]. This core comprises 119 proteins, including toxins such as phenol-soluble modulins, haemolysin, and leukocidin, invasion factors like elastin-binding protein, autolysin, and extracellular adherence protein, enzymes such as nuclease and proteases, surface antigens, and immunoglobulin-binding proteins [41]. Since then, SA-derived EVs have been implicated as an exogenous worsening factor of AD and other allergic diseases by mechanisms as increasing the expression of pro-inflammatory cytokines [27,35]. Certain components from SA-derived EVs have been directly associated with extracellular matrix degradation, bacterial antibiotic resistance, and bacterial invasion [42,43,44], providing strong evidence for their role in virulence. 

Among the many factors that stimulate the release of SA-derived EVs, one can cite the exposure to DNA-damaging agents or UV radiation as a relevant trigger [45]. Antibiotics such as the β-lactam flucloxacillin and ceftaroline are capable of stimulating their secretion through different pathways, such as DNA-damaging agents and antibiotics that induce signaling responses, initiating vesicle formation in lysogenic strains of *S. aureus*, or in a prophage-independent manner by affecting the peptidoglycan layer [45].

## 4. SA-Derived EVs and AD Pathophysiology

In 2011, Hong et al. [24] conducted the first study to propose the role of *S. aureus*-derived EV in AD pathophysiology. The authors reported that applying SA-derived EVs in tape-stripping mouse skin resulted in skin inflammation similar to that observed in AD patients, displaying a profile encompassed by (i) thickened epidermis, (ii) dermal infiltration of polymorphonuclear cells, (iii) an increase in Th1/Th17 inflammatory cytokines, and (iv) elevated serum IgE [24]. Importantly, these modifications were observed without the presence of live bacterial cells [24].

*S. aureus* colonization and infection evoke an immune response from the host by secreting pathogenic molecules and toxins (reviewed in Seiti Yamada Yoshikawa et al., 2019) [46]. Among these, we emphasize α-hemolysin, a cytotoxic protein secreted by *S. aureus* which induces cell death [47] on different cell types [48,49], including keratinocytes [23]. *S. aureus* from colonized AD skin releases greater amounts of α-hemolysin compared to *S. aureus* from the skin of healthy individuals. Furthermore, there are reports suggesting *S. aureus* α-hemolysin production is linked to AD severity [25].

An in vitro analysis revealed that EVs-associated α-hemolysin was more cytotoxic, and more effective to induce HaCaT keratinocytes death than soluble α-hemolysin, producing necrosis and inducing epidermal thickening and eosinophilic inflammation in the dermis [25,34], which could be explained by their better delivery to the host cell when encapsulated inside EVs. In addition, despite in vivo studies that indicate that soluble and EV-associated α-hemolysin induced skin barrier disruption and epidermal hyperplasia, an EV-associated toxin was the responsible for the induction of dermal infiltration [25].

Some *S. aureus* strains, such as ATCC14458 [24], a related cytotoxin producer, can release cytotoxins into the lumen of the EVs and induce their secretion to the extracellular space. Besides the assistance in the killing of the host cells by transferring these cytotoxic factors inside EVs, which favors their integration into the target cell cytoplasm [25], their association also shields the toxins from neutralization by the host immune system [24,25,50].

Mutually, in vitro and in vivo studies indicate that SA-derived EVs can upregulate pro-inflammatory mediators that elicit the Th17 response with augmented production of IgE, triggering AD-like inflammation [21,24]. Although AD is a disease ruled by type 2 immune responses (Th2, IL-4 and IL-13), IL-17 contributes to the worsening of the symptoms by enhancing IL-4 production from Th2 cells [51]. Moreover, SA-derived EVs induce the secretion of CXCL8 and TNF-α by primary human keratinocytes, the recruitment of neutrophils, and the formation of neutrophil extracellular traps, leading to better *S. aureus* skin colonization. The stimulation of CXCL8 is TLR2- and NFκB-dependent, and the induction level positively correlates with the membrane lipid and protein A in a similar quantity as those from pathogenic *S. aureus* [33]. 

SA-derived EVs can mediate inflammatory responses in AD pathogenesis; Kim et al. [30] examined the effect of these EVs on human dermal microvascular endothelial cells (HDMECs). HDMECs treated with SA-derived EVs increased the expression of cell adhesion molecules such as E-selectin, VCAM1, and ICAM1, and IL-6 with improved recruitment of monocytes in a TLR-4/NFκB-dependent signaling pathway [30]. All these findings together suggest that SA-derived EVs are key molecules in AD pathogenesis and are potential therapeutic targets. Figure 2 illustrates the role of SA-derived EVs in AD and some key biological effects are summarized in Table 2.

## 5. SA-Derived EVs and Microbiota in the Context of AD

Several studies have described alterations in the microbiome composition and Shannon’s microbial diversity index in AD skin, demonstrating a prevalence of *S. aureus* strains in flares [52,53,54]. Kim et al. [26] demonstrated the recovery of the reduced microbial diversity AD skin after treatment and detected SA-derived EVs in serum samples of AD patients, suggesting a novel host-microbiota crosstalk via microbe-derived EVs [20]. Likewise, a pilot study targeting the EV microbiome in the serum of AD patients identified biomarkers based on metagenomic analysis of serum-derived microbial EVs reinforcing that they may be a helpful tool for diagnosis, prognosis, and treatment prediction [31].

Besides *S. aureus*, commensal microorganisms on the skin also release EVs. For example, the EVs from the commensal yeast *Malassezia sympodialis*, associated with episodes of atopic dermatitis, can work as vessels for delivering small RNAs [55]. On the other hand, EV derived from skin commensals can protect against *S. aureus* colonization by conditioning the human skin to enhance innate defenses and by interfering with *S. aureus* keratinocyte attachment [33].

Recent research has highlighted how microbe-derived EVs can help to regulate immune responses in allergic diseases [20]. EVs derived from the probiotic *Lactobacillus plantarum* have demonstrated the ability to restore cell viability in keratinocytes and reduce levels of secreted IL-6 from both keratinocytes and macrophages in response to SA-derived EVs in vitro [29]. Moreover, *Lactobacillus plantarum*-derived EVs administered orally in a mouse model of AD induced with SA-derived EVs inhibited IL-4 production and its consequent skin inflammation in vivo, resulting in decreased epidermal thickness, albeit no changes in the eosinophilic infiltration [29]. These findings point out *L. plantarum*-derived EVs as a possible candidate for preventing skin inflammation [29,35]. 

Reports show the successful treatment of AD-like skin lesions induced by SA-derived EV inoculation in mice ears with monoterpenoid thymol. Both the topical application of thymol-treated SA-derived EVs or treatment with thymol after intact EV administration reduced AD inflammation, decreased epidermal/dermal thickness, attenuated the infiltration of eosinophils/mast cells, and reduced the expression of proinflammatory cytokine/chemokine genes [28]. Furthermore, thymol inhibited Th1, Th2, and Th17-mediated inflammatory responses as well as reduced IgG2a, mite-specific IgE, and total IgE serum levels [28].

*Staphylococcus epidermidis* dysbiosis is associated with AD disease, but in the opposite way to *S. aureus*. Byrd et al. [56] demonstrated in 2017 that *S. epidermidis* predominance is associated with less severe AD disease in some patients, while its lower abundance is associated with a more severe clinical presentation. In fact, EVs from skin commensal microbes, including *S. epidermidis*, *S. lugdunensis*, and *S. hominis* play a potentially protective role against *S. aureus* skin colonization by decreasing the adherence of *S. aureus* to keratinocytes [31]. Interestingly, a recent study has reported that EVs derived from *S. epidermidis* can reduce the expression of proinflammatory genes (TNF-α, IL-1β, IL-6, IL-8, and iNOS) in MC903-treated HaCaT cells [36]. Moreover, *S. epidermidis* EVs induced the expression of human β-defensins and promoted epidermal cell renewal in HaCaT cells, in addition to suppressing proinflammatory genes [36]. Finally, in vivo studies showed that the topical administration of *S. epidermidis* EVs in an AD-like dermatitis model attenuated lesion formation and reduced AD-like inflammatory features, such as epidermal thickness, dryness, transepidermal water loss, pruritus, epidermal cell infiltration (CD4+ T cells and Gr1+ cells), Th2 cytokine gene expression (IL-4, IL-13 and TLSP), and IgE levels [36].

## 6. Future Research Questions

In this review, we describe an updated panorama of the research regarding SA-derived EVs and AD. Even though the subject is still in its initial approach, we believe that SA-derived EVs may have a role as novel players in AD pathogenesis. Overall, EVs are essential virulence factors for *S. aureus*, highlighted by the higher toxicity of the pathogen toxins observed when associated with/carried within EVs and their ability to disrupt the host immune system in favor of skin colonization.

Kobiela et al. [57] demonstrated that supernatants from *S. aureus* cultures hijack the EV secretion machinery of keratinocytes, increasing the content of filaggrin eliminated inside these vesicles. This would deplete the keratinocyte reservoir of this protein, which could possibly impair the antimicrobial functions of the epidermal layer, favoring bacteria colonization. Since the authors did not identify the component(s) responsible for this activity, we may hypothesize that SA-derived EVs in those supernatants could be involved in delivering the bacterial factors to the keratinocytes. Indeed, the proportion of factors/toxins that *S. aureus* might release in the free-form and in an EV-encapsulated manner remains to be explored. It would be even more interesting to assess whether AD-associated *S. aureus* strains are more prone to EV production than the ones colonizing healthy individuals, which could help to explain the dysbiosis observed in AD patients.

Although not the focus of this review, it is relevant to point out that the host has its own source of EVs, which exert a role in the shaping of the host immune profile, therefore counterbalancing the AD polarization driven by SA-derived EVs [58]. Cho et al. [59] showed that EVs from adipose tissue-derived mesenchymal stem cells ameliorate AD symptoms in an experimental model through undetermined effector mechanism(s). Furthermore, IFN-γ-primed mesenchymal stem cells secrete EVs that dampen AD symptoms by repressing Th2 responses [60]. Therefore, the phenotype that prevails would be determined by the relative contributions of SA-derived versus host-derived EVs. We still have a poor understanding of if and how the EV profile changes between atopic and non-atopic individuals; one can speculate that AD patients could even display different EV kinetics between resting and flare states. Thus, whether an “atopy-prone EV profile” can be delineated is an open question for future investigations.

The search for EV markers of AD would be very beneficial for early diagnosis, disease monitoring, or even therapeutic targets. One important advantage is the easy accessibility of EV samples from the blood. Indeed, some authors propose the use of blood fluids as a source of EVs with the aim of improving and facilitating AD diagnosis [26]. Although plasma and serum are the main sources for EV studies due to their abundance in liquid biopsies representing a low-invasive harvesting procedure [61], they include a diverse mixture of EVs from different cell types and tissues, but also from different species [62,63,64]. This might not be a concern in a laboratory routine, but some limitations are present when considering research purposes. Fortunately, the discrimination of the source of EVs seems to be technically possible, as exemplified by Yang et al. (2020) [31], who assessed the diversity of microbial EVs in the serum of AD patients by a metagenomic approach to establish a diagnostic model. Great advances in the use of mass spectrometry and computational biology should harbor the answer for faster and more detailed EV profiling in the context of patient screening [65].

Finally, it is important to keep in mind the bidirectional nature of this subject. Despite the great focus on the ways that SA-derived EVs can affect the host to benefit pathogen establishment, one might wonder whether host EVs could not affect *S. aureus* behavior. Recently, Wang et al. [66] demonstrated that the model plant *Arabidopis thaliana* uses its EVs to deliver mRNAs to the fungal pathogen *Botrytis cinerea*, attenuating its virulence. Although this idea remains largely unappreciated in the AD field, it opens new venues of investigation for exploring host—*S. aureus* interaction.

Among the many possibilities in the field of SA-derived EVs in atopic dermatitis, we proposed some drivers for a better understanding of AD, including its pathogenesis, identifying new biomarkers and targets for clinical intervention.

Is it possible to establish an AD-linked, SA-derived EV profile?

-Are AD-associated *S. aureus* strains related to different EV features?-Is the EV production machinery a viable target for targeted pharmacological intervention?

## 7. Conclusions

Microbe-derived EVs have various functions linked to cell-to-cell communication and are currently potential tools for diagnostic and therapeutic purposes. Microbe-derived EV studies show novel approaches for diseases such as atopic dermatitis. Future studies will elucidate gaps in the understanding of host and microbiota crosstalk, further facilitating novel diagnostic tools and therapeutic targets.

## Figures and Tables

**Figure 1 microorganisms-12-00531-f001:**
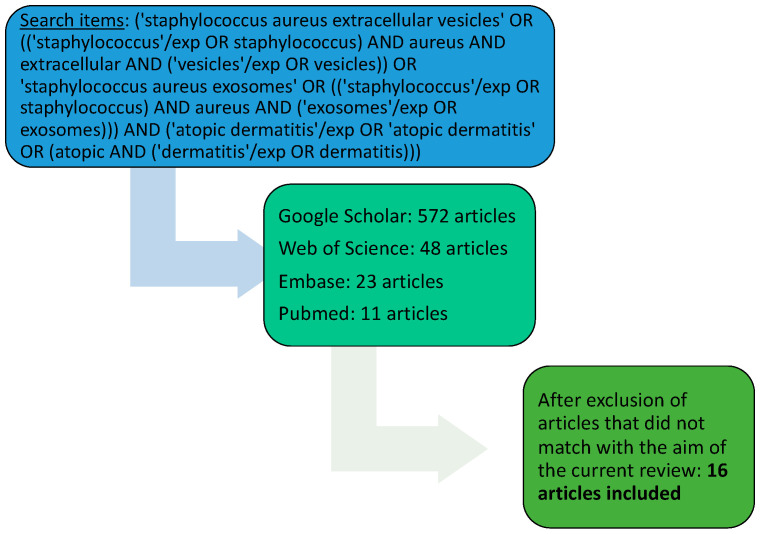
Review search terms. Graphical strategy and search results. The applied keywords and the main results of the search.

**Figure 2 microorganisms-12-00531-f002:**
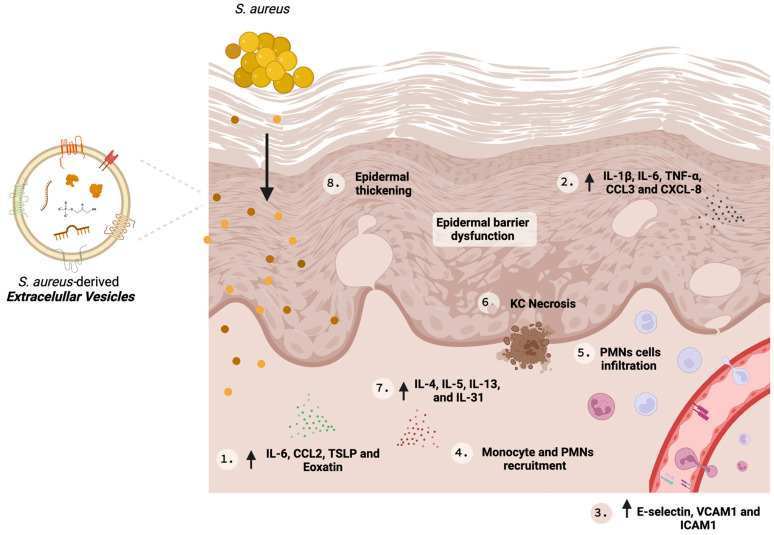
*S. Aureus*-derived EVs in AD pathogenesis. The effects of *S. aureus*-derived extracellular vesicles on skin cells are diverse. They can induce fibroblasts to secrete IL-6, CCL2, TSLP, and Eotaxin (1), and keratinocytes (KC) to secrete CXCL-8, IL-6, IL-1β, and TNF-α (2), all of which are proinflammatory mediators. Dermal microvascular endothelial cells upregulate the expression of the adhesion molecules E-selectin, VCAM1, and ICAM1 (3), which contribute to leukocyte migration (4). Secretion of CXCL-8 by KC also contributes to the cell migration, resulting in an increased polymorphonuclear cell (PMN) infiltration (5). Additionally, *S. aureus*-derived EVs can have a cytotoxic effect on KC, culminating in necrosis and skin barrier disruption (6). All these events together lead to eosinophilic inflammation in the dermis (7), epidermal hyperplasia (8), upregulation of proinflammatory mediators that elicit the Th2/Th17/Th1 response and augmented production of IgE. ↑: increased expression. Figure created with BioRender.com (accessed on 9 February 2024).

**Table 1 microorganisms-12-00531-t001:** Milestones in SA-derived EVs and AD research.

Year	Main Findings	Reference
2009	First description that *S. aureus* spontaneously producing EVs	Lee et al. [23]
2011	SA-derived EVs induce AD-like inflammation in the skin and should be considered as novel diagnostic and therapeutic target for the control of AD	Hong et al. [24]
2014	EV-associated α-hemolysin induces necrosis of keratinocytes, skin barrier disruption and epidermal hyperplasia	Hong et al. [25]
2017	Metagenomic analysis together with serum detection of pathogen-specific EVs provides a model for identification and diagnosis of pathogens of AD	Kim et al. [26]
2017	SA-derived EVs as potent mediator for exacerbation of AD severity	Jun et al. [27]
2018	Thymol disrupts SA-derived EVs and suppresses inflammatory responses in AD-like skin lesions aggravated by *S. aureus* EVs	Kwon et al. [28]
2018	*L. plantarum*-derived EVs might help prevent skin inflammation	Kim et al. [29]
2019	SA-derived EVs as proinflammatory factors could mediate immune cell infiltration in AD by efficiently inducing endothelial cell activation and monocyte recruitment	Kim et al. [30]
2020	Bacterial EVs carry several types of molecules: proteins, glycoproteins, mRNAs and small RNA species, as mammalian EVs do, but also carbohydrates	Dagnelie et al. [20]
2020	A pilot study indicates microbial EVs as potential biomarkers for AD diagnosis	Yang et al. [31]
2020	EVs from Gram-positive bacteria carry a diversity of cargo compounds that have a role in bacterial competition, survival, invasion, host immune evasion, and infection	Bose et al. [21]
2021	EVs represent a novel *S. aureus* secretory system that is affected by a variety of stress responses and allows the delivery of biologically active pore-forming toxins and other virulence determinants to host cells	Wang et al. [32]
2021	EVs of *S. aureus* strains from the lesional skin of AD patients show an enhanced membrane lipid and protein A content compared to the strains from the non-lesional sites with enhanced proinflammatory potential	Staudenmaier et al. [33]
2022	Microbial EV therapy may offer a variety of benefits over live biotherapeutics and human cell EV (or exosome) therapy for the treatment of intractable diseases	Yang et al. [34]
2023	Bacterial EVs may exert diverse effects on immune responses both beneficial or pathogenic role in patients with allergic and immunologic diseases	Choi et al. [35]
2023	SA-derived EVs reduced AD-like skin inflammation in mice and may potentially be a bioactive nanocarrier for the treatment of AD	Zhou et al. [36]

AD: atopic dermatitis; EVs: extracellular vesicles; *S. aureus*: *Staphylococcus aureus*; *L. plantarum*: *Lactobacillus plantarum*.

**Table 2 microorganisms-12-00531-t002:** Biological effects of SA-derived EVs.

*S. aureus* Strain	Study Type	Experimental Model	Inflammatory Effector Molecules Upregulated	Histological Features	Others Observed Effects	Ref.
ATCC14458	In vivo	EV were applied by tape stripping mouse skin	IL-4, IL-5, IL-17, IFN-γ	Infiltration of polymorphonuclear cells and epidermal thickness	-	[24]
03ST17	In vivo	Topical application of EVs into DFE induced lesions on AD-like mouse model;	IL-13, IL-31, CXCL8, CCL2 and CCL3	Infiltration of polymorphonuclear cells and epidermal thickness	Severe eczematous dermatitis, swelling, redness, bullae, and eschar formation	[27]
USA300	In vitro	Primary human keratinocytes	CXCL8 and TNF-α	-	Recruitment of neutrophils and induction of NETs	[33]
ATCC 6538	In vitro	Immortalized human dermal microvascular endothelial cells	E-selectin, VCAM1, ICAM1 and IL-6	-	Recruitment of monocytes	[30]
ATCC14458 and from AD patients	In vitro	Immortalized human keratinocytes	IL-1β and IL-6	-	Cytotoxic effect associated with EV-α-Hemolysin	[25]
ATCC14458	In vitro	Primary mouse dermal fibroblasts	IL-6, TSLP, CCL2 and Eotaxin	-	-	[24]
03ST17	In vitro	Immortalized human keratinocytes	IL-6	-	-	[27]

*S. aureus*: *Staphylococcus aureus*; AD: atopic dermatitis; EV: extracellular vesicles; NETs: neutrophil extracellular traps; Ref.: references; DFE: *Dermatophagoides farinae* extract.

## Data Availability

Data sharing not applicable.

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
