# Peer review of "State of the Art on the Role of Staphylococcus aureus Extracellular Vesicles in the Pathogenesis of Atopic Dermatitis"

_microorganisms, 2024, doi:10.3390/microorganisms12030531_

Round 1

Reviewer 1 Report

Comments and Suggestions for Authors

This is a review manuscript on the role of EV produced by S. aureus. The topic is interesting, but there are some concerns as follows.

1. Although this is a review paper, the style and organization of manuscript are similar to original article, i.e., including "Materials and Methods" and "Discussion". 

2. EV is a kind of cargo of nucleic acids, proteins, lipids, enzymes, as written in the manuscript. These components are included in the EV. In this manuscript, authors describe the biological effect of  EVs. However, there are less descriptions about those contents in EVs (sort of substances, relative volume of them, etc.). And also the stability of EVs is not shown. Probably, EVs will be broken after production from bacterial cells, and the content, such as toxin proteins are released. In such case, it is not clear whether the observed biological effects are caused by the particle of "EV" itself, or by the proteins (toxins) included in the EVs. Because there have been lots of review papers on S. aureus toxins, if the biological effects described in the manuscript are due to S. aureus toxins and other substances, the value of this manuscript appears to be substantially low, showing no novel information.   

Comments on the Quality of English Language

Almost OK.

Author Response

Reviewer Comments:

Reviewer 1

This is a review manuscript on the role of EV produced by S. aureus. The topic is interesting, but there are some concerns as follows.

  1. Although this is a review paper, the style and organization of manuscript are similar to original article, i.e., including "Materials and Methods" and "Discussion". 

Answer: Thank you for your comments. We reorganized the manuscript in a better display as a review article. The changes are highlighted in blue in the R1-manuscript.

  1. EV is a kind of cargo of nucleic acids, proteins, lipids, enzymes, as written in the manuscript. These components are included in the EV. In this manuscript, authors describe the biological effect of  EVs. However, there are less descriptions about those contents in EVs (sort of substances, relative volume of them, etc.). And also the stability of EVs is not shown. Probably, EVs will be broken after production from bacterial cells, and the content, such as toxin proteins are released. In such case, it is not clear whether the observed biological effects are caused by the particle of "EV" itself, or by the proteins (toxins) included in the EVs. Because there have been lots of review papers on S. aureus toxins, if the biological effects described in the manuscript are due to S. aureus toxins and other substances, the value of this manuscript appears to be substantially low, showing no novel information.   

Answer: Thank you for your comments. We agree with your concerns regarding EVs produced by bacteria, and we think that this is a novel approach to increase the understanding of how S. aureus remains the main challenge in atopic dermatitis progression. We agree that there are many studies discussing S. aureus toxins and enterotoxins, including studies from our group, but we think that this new approach of SA-derived EVs can contribute to open horizons for further questions that remains unclear, as we wrote in our conclusions: “Microbes-derived EVs have been shown to have various functions linked to cell-to-cell communication and are currently being considered for their diagnostic and therapeutic purposes. Microbes-derived EVs studies are the beginning of a novel approach of investigations, but further studies are still needed to elucidate the unique host and microbiota crosstalk temporal, to support the development of disease biomarkers and therapeutic targets”, and as we  addressed in the topic “Future research questions”. Our group is actually conducting a research exploiting the duality of EVs from AD patients and the cargo of S. aureus content (data not published).

Although the stability of EVs could be a valid concern, it is unlikely that EVs would spontaneously break after being produced. Mechanistically, the function of EVs is to allow the long-distance delivery of sensible components within a safe vessel. Therefore, they are known to be very stable structures, as attested by several technical papers that showed that EVs can resist different methods of manipulation and storage for long periods of time (e.g. doi: 10.1002/jev2.12238; doi: 10.1080/10717544.2020.1869866; doi: 10.1248/bpb.b16-00891). In addition, in accordance with the guidelines for EVs studies settled by the ISEV community (most recent version available in doi: 10.1002/jev2.12404.), it is mandatory that any study about EVs present data on the identity and the integrity of the material (identification markers, particle counting, zeta potential, electron microscopy, etc) to guarantee that the work do employ EVs. Hence, we findings cited in our manuscript do follow quality and scientific standards relevant for the topic.

Reviewer 2 Report

Comments and Suggestions for Authors

General evaluation:  aim, novelty, and significance

This review article aims to examine the role of Staphylococcus aureus secretes extracellular vesicles (SA-derived EVs) in Atopic dermatitis (AD) physiopathology and progression and to explore the potential crosstalk between host and microbiota.

Outlining and defining the AD physiopathology and progression in view of the recent discoveries and publications may give a comprehensive image of the state of the art of this topic and add valuable information to the field-interested people. 

Major required changes

·       The article structure is simulating the format of the research article as divided into (Introduction, Materials and Methods, results then discussion). This is not usual in review articles. It may be innovative but at the same time confusing and misleading. For example the reader may think that the section (results) may refer to results outputting from the current work and this not the fact. In such case it be referred as data and not results since the information is coming from previous research works.

·       So, the sections should be redesigned to fulfil the article objectives.

·        The section (2. Materials and Methods) should be more expressing and it is suggested to be (2. Methodologies) as this review paper is not the same as the research article. For example this section in this review does not include materials).

·        Under the section(2. Materials and Methods) there is only one subsection (2.1. Review strategy). There is no point of subdividing one section into one subsection.

·        The numbering of the section (3.2. SA-derived EVs and microbiota in the context of AD)should be corrected to (3.3.)

·       Figure 2 title (Figure 2. S. Aureus-derived EVs in AD pathogenesis.) is merged with the following text . So please separate them.

·       Change the title (5. Future Directions) to (5. Future research questions).

·       The article needs a careful and attentive major revision.

Minor required linguistic corrections

v Abstract

For better clarity, modify the sentence (Extracellular vesicles … cells.) into (Extracellular vesicles mediate cell-to-cell communication by transporting and delivering bioactive molecules such as nucleic acids, proteins, and enzymes to recipient cells.)

Modify the sentence (Staphylococcus …. layers.) into (Staphylococcus aureus spontaneously secretes extracellular vesicles (SA-derived EVs), which spread throughout the skin layers.)

Change (models; in) to (models, in)

Change (in addition to been) to (in addition to being)

v Introduction

Change (whose prevalence are approximately) to (whose prevalence is approximately).

Change (associated to AD development) to (associated with AD development).

What is meant by the word (crescent) in this context (crescent attention [11]).??

Change (to clinical evidences) to (to clinical evidence).

Change (and originated from) to (, originating from)

Change (Despite these categorization) to (Despite this categorization)

Change (impacting on the physiopathology) to (impacting the physiopathology)

v Materials and methods

Change (between 2009 to 2023.) to (between 2009 and 2023.)

Change (did not match with the purpose) to (did not match the purpose)

v Results

Change (bacteria outer membrane layer) to (bacterial outer membrane layer)

Change (possess a outer membrane) to (possess an outer membrane).

Change (being named as membrane vesicles) to (being named membrane vesicles).

The following sentence should be released from the brackets ([however, along this review we will keep the term “extracellular vesicles/EVs” regardless of the source organism].) and be an independent sentence.

Change (along this review we) to (in this review, we).

Change (Lee et al) to (Lee et al.)

Change (another group) to (another research group)

Change (This core is comprised of 119) to (This core comprises 119).

Change (immunoglobulin-binding protein) to (immunoglobulin-binding proteins).

Change (associated to extracellular) to (associated with extracellular).

Change (induce a signaling responses to induce vesicle formation) to (induce signaling responses, initiating vesicle formation).

Change (to the observed in AD) to (to that observed in AD).

Change (Importantly these modifications) to (Importantly, these modifications).

Change (endothelial cells to upregulate) to (endothelial cells upregulate).

Change (and disruption of the skin barrier) to (and skin barrier disruption)

Change (associated to episodes) to (associated with episodes)

Change (in decrease epidermal) to (in decreased epidermal).

Change (thickness albeit) to (thickness, albeit).

Change (pro-inflammatory) to (proinflammatory).

Change (with more severe) to (with a  more severe)

Change (to potentially play a protective role) to (to play a potentially protective role)

Change (associated to) to (associated with).

Change (favor the skin colonization.) to (favor skin colonization.)

v Discussion

Change (by relative) to (by the relative)

Change (the main source) to (the main sources)

Change (that essence) to  (that the essence)

v Conclusions

Change (have shown to) to (have been shown to)

Comments on the Quality of English Language

Attentive revision needed to filtrate the article from the linguistic errors.

Author Response

Reviewer 2

This review article aims to examine the role of Staphylococcus aureus secretes extracellular vesicles (SA-derived EVs) in Atopic dermatitis (AD) physiopathology and progression and to explore the potential crosstalk between host and microbiota.

Outlining and defining the AD physiopathology and progression in view of the recent discoveries and publications may give a comprehensive image of the state of the art of this topic and add valuable information to the field-interested people.  

Answer: We would like to thank Reviewer #2 for the careful reading of our manuscript and for the additional comments and corrections. The changes are highlighted in red in the manuscript and the answers are below described.

Major required changes

  • The article structure is simulating the format of the research article as divided into (Introduction, Materials and Methods, results then discussion). This is not usual in review articles. It may be innovative but at the same time confusing and misleading. For example the reader may think that the section (results) may refer to results outputting from the current work and this not the fact. In such case it be referred as data and not results since the information is coming from previous research works. 
  • So, the sections should be redesigned to fulfil the article objectives.

Answer: We reorganized the manuscript in a better display as a review article.

  • The section (2. Materials and Methods) should be more expressing and it is suggested to be (2. Methodologies) as this review paper is not the same as the research article. For example this section in this review does not include materials).

Answer: We reorganized the manuscript, excluding Materials and Methods in a better display as a review article. We just kept “Review Strategy”.

  • Under the section(2. Materials and Methods) there is only one subsection (2.1. Review strategy). There is no point of subdividing one section into one subsection.

Answer: We reorganized the manuscript in a better display as a review article and excluded subsection 2.1.

  • The numbering of the section (3.2. SA-derived EVs and microbiota in the context of AD)should be corrected to (3.3.)

Answer: We reorganized the manuscript and section 3.2 is now section 4.

  • Figure 2 title (Figure 2. S. Aureus-derived EVs in AD pathogenesis.) is merged with the following text . So please separate them.

Answer: The title of Figure 2 is merged with the caption of the figure following format of the journal template.

  • Change the title (5. Future Directions) to (5. Future research questions).

Answer: Thank you for your suggestion, we changed the title as requested. (section 6. Future Research Questions in the revised version)

  • The article needs a careful and attentive major revision.

Answer: We would like to thank Reviewer #2 for the careful reading of our manuscript and for the additional comments and corrections. We have addressed your concerns, and we believe this has greatly improved the quality of the manuscript. We hope our manuscript merits publication now in its revised form and we hope that the undertaken changes will be satisfactorily.

Minor required linguistic corrections

Abstract

For better clarity, modify the sentence (Extracellular vesicles … cells.) into (Extracellular vesicles mediate cell-to-cell communication by transporting and delivering bioactive molecules such as nucleic acids, proteins, and enzymes to recipient cells.)

Modify the sentence (Staphylococcus …. layers.) into (Staphylococcus aureus spontaneously secretes extracellular vesicles (SA-derived EVs), which spread throughout the skin layers.)

Change (models; in) to (models, in)

Change (in addition to been) to (in addition to being)

Answer: Thank you for your suggestions, we made all changes as requested.

Introduction

Change (whose prevalence are approximately) to (whose prevalence is approximately).

Change (associated to AD development) to (associated with AD development).

What is meant by the word (crescent) in this context (crescent attention [11]).??

Change (to clinical evidences) to (to clinical evidence).

Change (and originated from) to (, originating from)

Change (Despite these categorization) to (Despite this categorization)

Change (impacting on the physiopathology) to (impacting the physiopathology)

Answer: Thank you for your suggestions, we made all changes as requested. We replaced “crescent attention“ by “gathering increased attention”, that sounds better.

Materials and methods

Change (between 2009 to 2023.) to (between 2009 and 2023.)

Change (did not match with the purpose) to (did not match the purpose)

Answer: Thank you for your suggestions, we made all changes as requested.

Results

Change (bacteria outer membrane layer) to (bacterial outer membrane layer)

Change (possess a outer membrane) to (possess an outer membrane).

Change (being named as membrane vesicles) to (being named membrane vesicles).

The following sentence should be released from the brackets ([however, along this review we will keep the term “extracellular vesicles/EVs” regardless of the source organism].) and be an independent sentence.

Change (along this review we) to (in this review, we).

Change (Lee et al) to (Lee et al.)

Change (another group) to (another research group)

Change (This core is comprised of 119) to (This core comprises 119).

Change (immunoglobulin-binding protein) to (immunoglobulin-binding proteins).

Change (associated to extracellular) to (associated with extracellular).

Change (induce a signaling responses to induce vesicle formation) to (induce signaling responses, initiating vesicle formation).

Change (to the observed in AD) to (to that observed in AD).

Change (Importantly these modifications) to (Importantly, these modifications).

Change (endothelial cells to upregulate) to (endothelial cells upregulate).

Change (and disruption of the skin barrier) to (and skin barrier disruption)

Change (associated to episodes) to (associated with episodes)

Change (in decrease epidermal) to (in decreased epidermal).

Change (thickness albeit) to (thickness, albeit).

Change (pro-inflammatory) to (proinflammatory).

Change (with more severe) to (with a  more severe)

Change (to potentially play a protective role) to (to play a potentially protective role)

Change (associated to) to (associated with).

Change (favor the skin colonization.) to (favor skin colonization.)

Answer: Thank you for your suggestions, we made all changes as requested.

Discussion

Change (by relative) to (by the relative)

Change (the main source) to (the main sources)

Change (that essence) to  (that the essence)

Answer: Thank you for your suggestions, we made all changes as requested.

Conclusions

Change (have shown to) to (have been shown to)

Answer: Thank you for your suggestions, we made all changes as requested.

Comments on the Quality of English Language

Attentive revision needed to filtrate the article from the linguistic errors.

Answer: Thank you for your careful review. We provided an English native speaker for reviewing our paper and we hope that it reached your content.

Reviewer 3 Report

Comments and Suggestions for Authors

The manuscript regarding the topic of EVs is well-completed. However, as the type of submission is a review, the presentation showed is not the best, due to is not a "original article" the structure is different. It is more practical by titles of the matter that the authors present. For example, material and methods do not correspond here. If the authors want to explain how was the way to revise the literature of the review, they should indicate it as supplementary material. The authors should eliminate titles "2 and Figure 1"; the title results should not appear either, and the titles should be changed to the important points of the revision that the authors want to show. In Figure 2, revise the bacteria name and write it in italics and lowercase. Finally, it is recommended to extend or add more titles to the revision in order to explain the topic presented with more detail and probably think of another title for the review.

One interesting concern about the revision that should be important to explain is the alpha-hemolysin, due to S. aureus being characterised by producing beta hemolysin and not the other; how can the authors explain this?

Comments on the Quality of English Language

In general, revise minimal mistakes in the writing, for example, "in addition to been implicated as an exogenous worsening factor for the disease." The verb should be "being". 

I recommend the revision of the whole manuscript in order to improve it.

Author Response

Reviewer Comments:
Reviewer 3

 The manuscript regarding the topic of EVs is well-completed. However, as the type of submission is a review, the presentation showed is not the best, due to is not a "original article" the structure is different. It is more practical by titles of the matter that the authors present. For example, material and methods do not correspond here. If the authors want to explain how was the way to revise the literature of the review, they should indicate it as supplementary material. The authors should eliminate titles "2 and Figure 1"; the title results should not appear either, and the titles should be changed to the important points of the revision that the authors want to show. In Figure 2, revise the bacteria name and write it in italics and lowercase. Finally, it is recommended to extend or add more titles to the revision in order to explain the topic presented with more detail and probably think of another title for the review.

Answer: We want to thank Reviewer 3 for your comments. We reorganized the manuscript in a better display as a review article. The changes are highlighted in blue in the R1-manuscript. We hope our manuscript merits publication now in its revised form and we hope that the undertaken changes will be satisfactorily.

One interesting concern about the revision that should be important to explain is the alpha-hemolysin, due to S. aureus being characterized by producing beta hemolysin and not the other; how can the authors explain this?

Answer: Thank you for your comment. S. aureus secrete numerous exotoxins, including a group of polypeptides capable of damaging the host cell plasma membrane. These polypeptides include pore-forming toxins [PFT: α-hemolysin and the bi-component leukocidins γ-hemolysin, the Panton Valentine leukocidin (PVL), LukED, and LukGH/AB], β-hemolysin (a neutral sphingomyelinase), and the phenol soluble modulins (PSMs, a family of small amphipathic peptides) (doi: 10.3389/fcimb.2012.00012). It has been described that α-hemolysin is the most characterized virulence factor of S. aureus, and that in contrast to α-Hemolysin,  β-hemolysin lysis of red blood cells is only observed after the cells are switched to low temperature, suggesting that the lytic activity of β-hemolysin is not as efficient as that of other hemolysins (doi: 10.3390/toxins10060252). We included these references in the R1-manuscript.

Comments on the Quality of English Language

In general, revise minimal mistakes in the writing, for example, "in addition to been implicated as an exogenous worsening factor for the disease." The verb should be "being". 

I recommend the revision of the whole manuscript in order to improve it.

Answer: Thank you for your inputs. We provided an English native speaker for reviewing our manuscript.

Round 2

Reviewer 1 Report

Comments and Suggestions for Authors

None

Reviewer 2 Report

Comments and Suggestions for Authors

The authors responded appropriately to the raised comments. 

L299-300, Change “Future studies will elucidate gaps in the understanding of host and microbiota crosstalk,” to (Future studies will elucidate gaps in understanding host and microbiota crosstalk,”

Comments on the Quality of English Language

Ordinary English revision and editing is required.

Reviewer 3 Report

Comments and Suggestions for Authors

The authors made significant changes to the manuscript, except for Figure 1 and Section 2, which could use improvement. Their effort to enhance the previous version is appreciated.